# 'Missingness' in health care: Associations between hospital utilization and missed appointments in general practice. A retrospective cohort study

**Andrea E. Williamson**[1]*, **Ross McQueenie**[2], **David A. Ellis**[3], **Alex McConnachie**[4], **Philip Wilson**[5]

1 General Practice and Primary Care, School of Medicine, Dentistry and Nursing, MVLS, University of Glasgow, Glasgow, United Kingdom, 2 Public Health Scotland, NHS Scotland, Glasgow, United Kingdom, 3 School of Management, University of Bath, Bath, United Kingdom, 4 Robertson Centre for Biostatistics, Institute of Health and Wellbeing, MVLS, University of Glasgow, Glasgow, United Kingdom, 5 Centre for Rural Health, Institute of Applied Health Sciences, University of Aberdeen, Aberdeen, United Kingdom

* andrea.williamson@glasgow.ac.uk

**Data Availability Statement:** These data were available from NHS Scotland. Permission for access was granted to the study team only from

## Abstract

### Objectives

Are multiple missed appointments in general practice associated with increased use of hospital services and missingness from hospital care? This novel study explores this in a population representative sample for the first time.

### Design, setting, participants

A large, retrospective cohort (*n* = 824,374) of patient records from a nationally representative sample of GP practices, Scotland, 2013–2016. Requested data were extracted by a Trusted Third Party for the NHS, anonymised and linked to a unique patient ID, in the NHS Safehaven for analysis using established NHS Scotland linkage. We calculated the per-patient number of GP missed appointments from individual appointments and investigated the likelihood of hospital appointment or admission outcomes using a negative binomial model offset by number of GP appointments made. These models also controlled for age, sex, Scottish Index of Multiple Deprivation (SIMD) and number of long- term conditions (LTCs).

### Main outcome measures

*Hospital attendance*: Outpatient clinic attendances; hospital admissions; Emergency Department (ED) attendances. *Hospital missingness*: 'Did not attend' (DNAs) outpatient clinic appointments, 'irregular discharges' from admissions, and 'left before care completed' ED care episodes.

### Results

*Attendance*: Patients in the high missed GP appointment (HMA) category were higher users of outpatient services (rate ratio (RR) 1.90, 95% confidence intervals (CI) 1.88–1.93)

the participating GP practices and the Public Benefit and Privacy Panel NHS Scotland. Requests to access these data in the same manner as the authors can be made to http://www.escro.co.uk/ for general practice data and to https://www.isdsscotland.org/Products-and-Services/eDRIS/ to host the analysis of general practice data and for permissions and access to secondary care data. The authors did not have any special access privileges that other researchers would not have. Analysis code is available from the authors upon reasonable request. The code will replicate the study findings, but the analysis could also be replicated using other statistical software.

**Funding:** AEW DAE AMcC and PW received funding for this research from a Scottish Government Chief Scientist Office research grant (CZH/4/41118) https://www.cso.scot.nhs.uk/ with Safe Haven and data linkage costs supported in lieu by the DSLS at Scottish Government. The funders had no role in study design, data collection and analysis, decision to publish, or preparation of the manuscript.

**Competing interests:** The authors have declared that no competing interests exist.

compared to those who missed none (NMA) with a much higher attendance risk at mental health services (RR 4.56, 95% CI 4.31–4.83). HMA patients were more likely to be admitted to hospital; general admissions (RR 1.67, 95% CI 1.65–1.68), maternity (RR 1.24, 95% CI 1.20–1.28) and mental health (RR 1.23, 95% CI 1.15–1.31), compared to NMA patients. Missing GP appointments was not associated with ED attendance; (RR 1.00, CI 0.99–1.01).

*Missingness*: HMA patients were at greater risk of missing outpatient appointments (RR 1.62, 95% CI 1.60–1.64) than NMA patients; with a much higher risk of non-attendance at mental health services (RR 7.83, 95% CI 7.35–8.35). Patients were more likely to leave hospital before their care plan was completed-taking 'irregular discharges' (RR 4.56, 95% CI 4.31–4.81). HMA patients were no more at risk of leaving emergency departments 'without care being completed' (RR1.02, 95 CI 0.95–1.09).

## Conclusions

Patients who miss high numbers of GP appointments are higher users of outpatient and inpatient hospital care but not of emergency departments, signalling high treatment burden.

The pattern of 'missingness' is consistent from primary care to hospital care: patients who have patterns of missing GP appointments have patterns of missing many outpatient appointments and are more likely to experience 'irregular discharge' from in-patient care. Missingness from outpatient mental health services is very high.

Policymakers, health service planners and clinicians should consider the role and contribution of 'missingness' in health care to improving patient safety and care.

## Introduction

Health systems across the world regularly have to manage high demand. When this is happening, patients who miss health care appointments may appear to provide respite, but this 'missingness' could inadvertently increase health inequalities [1] and potentially increase the overall burden of morbidity. Our previous research about patient level patterns of missed appointments in general practice, which had high population coverage, found that repeatedly missing appointments is associated with greatly increased mortality risk, independent of both recorded morbidity and the total number of appointments made [2]. The relative increased risk associated with missing appointments is greatest among patients with mental health morbidities and without known physical morbidities. These patients died prematurely, most commonly from non-natural external factors such as suicide. The absolute risk of mortality attributable to missing general practice appointments is however greatest for those with known mental and physical morbidities. Around 5% of those who miss two appointments per year die within 12 months [3].

We make the distinction between single missed GP appointments and many, because with few exceptions], much of the existing literature concerning missed appointments tends to conflate the two [1]. This leads to conflicting evidence about what works to encourage attendance. For example, while a text message reminder may be an effective intervention for patients who miss a single appointment, this may provide little benefit for patients with complex needs.

We hypothesise that several potential mechanisms may explain the link between repeated missed appointments and mortality. One causal mechanism could be a failure to take up preventive care opportunities. For example, a lack of engagement with diabetes care has been associated with adverse outcomes [4]. Unsatisfactory use of the health care system will

therefore lead directly to harm. Alternatively, reverse causality might operate if missed appointments in general practice result from a high treatment burden [5]: frequent appointments for multiple conditions may lead to poor appointment adherence and subsequently higher use of unscheduled care [6].

Insight into the contribution of treatment burden (the work a patient has to do to manage their long term conditions) can now be obtained through the analysis of hospital service use, and whether the tendency for 'missingness' continues from general practice into hospital outpatient, inpatient or emergency department use.

This paper aims to investigate whether multiple missed appointments in general practice are associated with increased use of hospital services and with 'missingness' from these services. This may help target future interventions to improve attendance and reduce other aspects of 'missingness' in health care.

## Methods

### Study design

Almost all Scottish citizens are registered with a general practice which gate-keeps access to, receives summary information from, and prescribes on behalf of most secondary care services. A linked dataset can therefore yield valuable information on patterns of engagement across health services.

We used a large, retrospective cohort ($n$ = 824,374) of patient records from a nationally representative sample of GP practices across Scotland over a 3-year period. Requested data were extracted by a Trusted Third Party (TTP), Albasoft for the NHS, anonymised and linked to a unique patient ID in the Scottish NHS National Safehaven for analysis. A TTP is an independent organisation with the skills, infrastructure and permissions to safely extract and link data without having a direct interest in the use to which the data will be put. The TTP is required to ensure confidentiality for professionals and patients in the research process. Details of the analysis plan and data extraction are documented in existing publications [1,2,7].

Routinely available data for Scottish outpatient appointments, hospital admissions and Emergency Department attendance were linked to the GP dataset using established NHS Scotland linkage methods [8]. The descriptors for each linkage category are publicly available [9].

### Ethics approval

Letters of comfort were issued by the West of Scotland NHS Ethics Committee and the University of Glasgow College of Medical, Veterinary & Life Sciences Ethics Committee confirming that the full study did not need NHS ethics approval. Public Benefit and Privacy Panel approval for use of the data was granted by NHS Information Services Scotland in December 2016.

**Data analysis.** Patient and practice level GP appointment data used in this study were prepared as described previously to allow for straightforward comparisons [1,7]. GP appointments specifically refer to those that were attended face to face in a general practice setting delivered by members of a GP clinical team. Patients were then categorised into categories averaged over the three-year study period from 5 September 2013 until 5 September 2016: *zero missed appointments*: 0 over the 3 year period, *low missed appointments*: <1 per year average, *medium missed appointments*: 1–2 per year average, *high missed appointments*: >2 per year average GP appointments scheduled.

Long term conditions (LTC) data were produced using patients' primary care morbidity Read codes. LTC counts were generated using 43 long-term conditions and some patient prescription data as described by Barnett et al [10]. Codes were re-examined and refinement of

codes included in addiction/mental health categories were made. E.g. removing tobacco use codes from addictions categories Barnett et al's framework takes into account morbidity that is generally managed in primary care (such as migraine) and conditions that are more likely to be managed also in secondary care, such as cancer or stroke.

Data linkage was carried out using patient community health index (CHI) numbers–a unique identifier for each patient–from our general practice dataset to Scottish morbidity records (SMR). This allowed us to link GP appointment data with five other datasets: SMR00 (outpatient), SMR01 (general inpatient), SMR02 (maternity inpatient), and SMR04 (mental health inpatient and day case) as well as emergency department care records (ED). We further categorised SMR00 outpatient specialties into adult medical specialities, adult surgical specialties, maternity and reproductive health services, mental health services and paediatric services in order to provide more detailed analysis of outpatient data. *S1 File* Hospital Specialty Categories describes the specialties in each of these. Admissions and appointments from these datasets were included in the analysis from 1st September 2013 until 12th May 2017.

We used negative binomial modelling to examine the association between missed general practice appointments and attendance at out-patient clinics, hospital admissions, ED attendances, rates of missed hospital outpatient appointments, 'irregular discharges' following admission, and ED attendances where the patient 'left before care completed'. These models were offset by number of GP appointments made while also controlling for age, sex, Scottish Index of Multiple Deprivation (SIMD), and number of LTCs.

Scottish Index of Multiple Deprivation is the standard measure of socio-economic deprivation at the small area level (data zones) used in policy and research in Scotland. It includes measures relating to income, employment, education, health, access to services, crime and housing [11].

All statistical analyses were done in R software (version 3.4.0) [12].

We adopted a conservative criteria for statistical significance (P<0.01).

**Patient involvement.** Advice about the relevance and importance of this research was sought from the Royal College of General Practitioners Scotland Patient Participation in Practice (P3) Committee in 2016 to support the study team's application for data linkage of routine data sets. The P3 committee advised they agreed it was 'interesting, much needed and beneficial' and they have been updated on progress as the project has proceeded.

## Results

### Hospital care use

In order to ascertain whether patients with a high rate of missed appointments in general practice were also higher users of outpatient, general inpatient, (general, maternity and mental health services) and emergency department care, we first examined how the number of appointments at these services varied with missed appointment category, age, sex, SIMD level and number of long term conditions *(Table 1)*.

Patients in the high missed GP appointment category attended 2.5 times the number of outpatient appointments (mean 6.60, (standard deviation (SD) 9.39)) compared to those who missed no GP appointments (2.5 (SD 5.17)); and had more than five times the number of general hospital admissions (2.99 (SD 6.18)) as those who missed no GP appointments (0.53 (SD 2.04)). This effect was especially pronounced for maternity admissions at 19.5 times more likely; and was 8 times more likely for mental health services admissions. Other strong associations were with number of long term conditions and the patient's age.

In contrast, no factor examined appeared to be associated with ED attendance. The mean number of ED appointments was constant across primary care missed appointment categories.

**Table 1. Mean and standard deviation (SD) of number of attendances/admissions for outpatient (SMR00), general inpatient (SMR01), maternity inpatient (SMR02), mental health (SMR04) and Emergency Department (ED attendances).**

| | Mean (SD) hospital outpatient appointments attended (SMR00) | Mean (SD) general hospital admissions (SMR01) | Mean (SD) maternity hospital admissions (SMR02) | Mean (SD) mental health admissions (SMR04) | Mean (SD) emergency department attendances |
|---|---|---|---|---|---|
| **GP missed appointment category** | | | | | |
| Zero | 2.50 (5.17) | 0.53 (2.04) | 0.06 (0.43) | 0.01 (0.27) | 0.99 (2.24) |
| Low | 3.59 (5.17) | 0.94 (3.01) | 0.10 (0.59) | 0.02 (0.27) | 0.98 (2.24) |
| Med | 4.83 (7.75) | 1.56 (3.95) | 1.15 (0.78) | 0.04 (0.39) | 0.98 (2.34) |
| High | 6.60 (9.39) | 2.99 (6.18) | 1.17 (0.89) | 0.08 (0.74) | 0.98 (2.40) |
| **Age** | | | | | |
| 0–15 | 1.73 (4.11) | 0.47 (2.11) | 0.00 (0.05) | 0.00 (0.04) | 1.00 (2.31) |
| 16–30 | 2.56 (5.42) | 0.47 (1.92) | 0.23 (0.95) | 0.02 (0.36) | 0.99 (2.18) |
| 31–45 | 3.24 (6.49) | 0.61 (2.43) | 0.19 (0.79) | 0.03 (0.36) | 0.99 (2.31) |
| 46–60 | 3.57 (6.81) | 0.97 (3.17) | 0.00 (0.05) | 0.02 (0.29) | 0.98 (2.23) |
| 61–75 | 5.12 (7.59) | 1.64 (4.18) | 0.00 (0.00) | 0.02 (0.30) | 0.98 (2.22) |
| 76–90 | 6.50 (8.08) | 2.99 (5.50) | 0.00 (0.00) | 0.02 (0.29) | 0.98 (2.53) |
| 90+ | 4.41 (6.34) | 3.83 (4.94) | 0.00 (0.00) | 0.02 (0.19) | 0.98 (1.95) |
| **Sex** | | | | | |
| Male | 2.91 (6.24) | 0.93 (3.13) | 0.00 (0.00) | 0.02 (0.23) | 0.99 (2.24) |
| Female | 3.79 (6.24) | 0.93 (3.13) | 0.16 (0.77) | 0.02 (0.31) | 0.99 (2.28) |
| **Scottish Index of Multiple Deprivation (SIMD)** | | | | | |
| 1 | 3.47 (6.76) | 1.23 (3.77) | 0.12 (0.71) | 0.03 (0.35) | 0.98 (2.15) |
| 2 | 3.48 (6.47) | 1.14 (3.40) | 0.11 (0.68) | 0.02 (0.34) | 0.99 (2.53) |
| 3 | 3.61 (6.88) | 1.08 (3.50) | 0.09 (0.63) | 0.02 (0.41) | 0.97 (2.09) |
| 4 | 3.41 (6.61) | 1.01 (3.08) | 0.09 (0.56) | 0.02 (0.34) | 0.99 (2.30) |
| 5 | 3.46 (6,56) | 0.96 (3.22) | 0.09 (0.56) | 0.04 (0.34) | 0.99 (2.23) |
| 6 | 3.22 (6.00) | 0.83 (2.73) | 0.06 (0.42) | 0.04 (0.25) | 0.97 (2.04) |
| 7 | 3.32 (6,19) | 0.85 (2.91) | 0.07 (0.51) | 0.01 (0.25) | 0.98 (2.32) |
| 8 | 3.24 (6.10) | 0.84 (2.83) | 0.07 (0.48) | 0.01 (0.28) | 0.99 (2.18) |
| 9 | 3.31 (6.39) | 0.83 (3.00) | 0.07 (0.54) | 0.01 (0.19) | 1.00 (2.53) |
| 10 | 3.29 (6.22) | 0.73 (2.74) | 0.06 (0.42) | 0.01 (0.17) | 0.97 (2.04) |
| **Number of Long- term conditions** | | | | | |
| 0 | 1.78 (4.16) | 0.33 (1.34) | 0.09 (0.54) | 0.01 (0.14) | 0.99 (2.28) |
| 1–3 | 3.61 (6.47) | 0.90 (2.95) | 0.40 (0.64) | 0.02 (0.33) | 0.99 (2.28) |
| 4+ | 7.50 (9.36) | 2.97 (5.72) | 0.02 (0.33) | 0.05 (0.59) | 0.97 (2.31) |

Patients who missed no appointments had 0.99 (mean, (SD 2.24) accident and emergency services attendances during the follow up period, compared with 0.98 (SD 2.40) for patients in the highest missed appointments category.

We next used negative binomial regression modelling to examine associations between number of outpatient attendances, hospital admissions and ED attendances, and GP missed appointment category whilst controlling for age, sex, SIMD and number of long-term conditions, as shown in *Table 2*.

Patients in the high missed GP appointments category were higher users of outpatient services (rate ratio (RR) 1.90, 95% confidence intervals (CI) 1.88–1.93).

We then investigated associated rates of attendance by the 5 broad categories of outpatient clinics. *Table 3* shows that patients in the "high" missed GP appointment group had a significantly higher chance of attendance at all specialties. They showed an over two-fold higher risk

**Table 2. Negative binomial regression modelling showing association between GP missed appointment category and number of outpatient attendances or hospital admissions category.**

| | Relative risk of attendance or admission | | |
|---|---|---|---|
| | Rate ratio (95% CI), p-value | | |
| | Reference group: Zero missed GP appointments | | |
| | **Low** | **Medium** | **High** |
| Outpatient (SMR00) attendance | 1.29 (1.28–1.30) p<0.01 | 1.57 (1.56–1.59) p<0.01 | 1.90 (1.88–1.93) p<0.01 |
| General hospital (SMR01) admissions | 1.13 (1.12–1.14) p<0.01 | 1.28 (1.27–1.30) p<0.01 | 1.67 (1.65–1.68) p<0.01 |
| Maternity (SMR02) admissions | 1.04 (1.04–1.06) p<0.01 | 1.11 (1.08–1.14) p<0.01 | 1.24 (1.20–1.28) p<0.01 |
| Mental health (SMR04) admissions | 1.11 (1.04–1.18) p<0.01 | 1.09 (1.02–1.17) p<0.01 | 1.23 (1.15–1.31) p<0.01 |
| Emergency department (ED) attendance | 0.99 (0.98–0.99) p<0.01 | 0.99 (0.98–1.00) p = 0.01 | 1.00 (0.99–1.01) p = 0.90 |

Models show rate ratio (RR), 95% Confidence Intervals (CI) and control for age, sex, Scottish Index of Multiple Deprivation SIMD, and number of long-term conditions.

of attendance at adult medicine specialities (RR 2.02, 95% CI 1.98–2.06), a 53% increased risk of attendance at adult surgical specialities (RR 1.53, 95% CI 1.51–1.56), a two- fold increased attendance risk at maternity/reproductive health, an over 4.5 fold higher rate of attendance risk at mental health services (RR 4.56, 95% CI 4.31–4.83), and an over two-fold increase in attendance risk at paediatrics (RR 2.23, 95% CI 2.07–2.41) compared with those who missed no GP appointments.

Patients who had patterns of high missed GP appointments were also more likely to be admitted to hospital; general inpatient care (RR 1.67, 95% CI 1.65–1.68), maternity care (RR 1.24, 95% CI 1.20–1.28) and mental health care (RR 1.23, 95% CI 1.15–1.31), compared with patients who missed no primary care appointments.

There appeared to be no association between GP missed appointment status and attendance at ED: patients in the highest missed appointment category had no significantly increased risk

**Table 3. Negative binomial regression modelling showing association between primary care missed appointment category and number of appointments attended by speciality category.**

| | Relative risk of outpatient attendance by speciality category | | |
|---|---|---|---|
| | Rate ratio (95% CI), p-value | | |
| | Reference group: Zero GP missed appointments | | |
| | **Low** | **Medium** | **High** |
| Adult medicine attendance | 1.32 (1.31–1.34) p<0.01 | 1.64 (1.61–1.66) p<0.01 | 2.02 (1.98–2.06) p<0.01 |
| Adult surgical services attendance | 1.20 (1.19–1.21) p<0.01 | 1.36 (1.35–1.38) p<0.01 | 1.53 (1.51–1.56) p<0.01 |
| Maternity/reproductive health attendance | 1.40 (1.37–1.43) p<0.01 | 1.75 (1.69–1.81) p<0.01 | 2.06 (1.98–2.14) p<0.01 |
| Mental health services attendance | 1.86 (1.79–1.92) p<0.01 | 3.03 (2.90–3.18) p<0.01 | 4.56 (4.31–4.83) p<0.01 |
| Paediatrics attendance | 1.53 (1.48–1.59) p<0.01 | 1.96 (1.86–2.06) p<0.01 | 2.23 (2.07–2.41) p<0.01 |

Models show rate ratio (RR), 95% Confidence Intervals (CI) and control for age, sex, SIMD, and number of long-term conditions.

of ED attendance (RR 1.00, CI 0.99–1.01) compared with patients who missed no GP appointments.

*Fig 1* summarises hospital attendance and admission activity by missed GP appointments category.

### 'Missingness' from hospital care

In order to ascertain whether patients who missed multiple GP appointments were at greater risk of missing hospital outpatient appointments, experiencing 'irregular discharges' from inpatient care and who 'left before care completed' from emergency department care, we used negative binomial modelling to measure risk whilst controlling for demographic factors and number of LTCs as already described.

Patients with patterns of high missed GP appointments (HMA) were at greater risk of missing outpatient appointments (RR 1.62, 95% CI 1.60–1.64) than those who missed no GP appointments (NMA).

The analysis by the same five categories of outpatient specialities is shown in *Table 4*. Patients with HMA had an over three-fold risk of non-attendance at adult medicine specialities (RRR 3.80, 95% CI 3.68–3.93) and adult surgical specialities (RR 3.03, 95% CI 2.95–3.12), over 4.5 fold risk of non-attendance at maternity and reproductive health appointments (RR 4.64, 95% CI 4.36–4.94), nearly eight-fold risk of non-attendance at mental health services appointments (RR 7.83, 95% CI 7.35–8.35), and a tripled risk of non-attendance at paediatrics appointments (RR 3.22, 95% CI 2.86–3.63) compared with those with NMA.

Rates of 'irregular discharge' from inpatient hospital care are presented in *Table 5*. Patients were more likely to leave hospital before their care plan was completed-taking 'irregular discharges' for patients with HMA compared to NMA (RR 4.56, 95% CI 4.31–4.81). Binomial regression modelling was only possible for general admissions, due to low numbers for maternity and mental health care admissions.

Patients in the highest missed GP appointment group were no more at risk of leaving emergency departments 'without care being completed' (RR1.02, 95% CI 0.95–1.09) than patients who missed no GP appointments.

*Fig 2* summarises missingness from hospital care by GP missed appointment category.

## Discussion

### Statement of principal findings

Patients who miss high numbers of GP appointments are higher users of outpatient and inpatient hospital care but not of emergency departments.

The pattern of 'missingness' from care is consistent from primary care to hospital care: patients who have high patterns of missing GP appointments have patterns of missing many outpatient appointments and are more likely to experience 'irregular discharge' from inpatient care. Missingness from outpatient mental health services is particularly high.

Our findings do not support the hypothesis that patients who miss multiple GP appointments use ED services as a proxy. Whilst the risk of attendance at all scheduled secondary care services appeared to show a 'dose-based' increased risk alongside an increasing number of missed GP appointments, there was no association between the level of missed GP appointments and ED attendance. Similarly, whilst increased levels of GP missed appointments were associated with a 'dose-based' increased risk of not attending scheduled secondary care services, there was no increased risk of not waiting to be seen in ED for any level of missed GP appointment category.

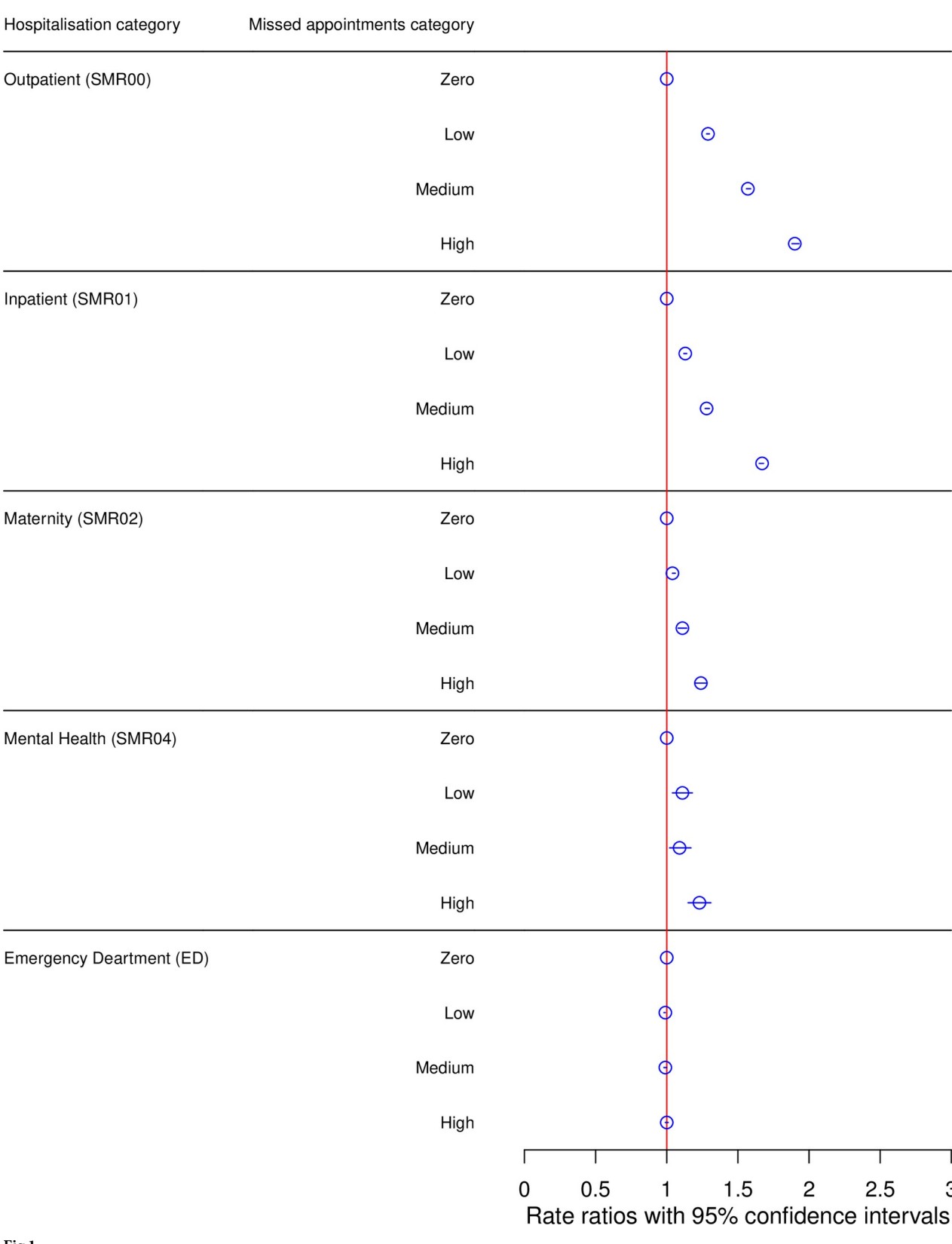

**Fig 1.**

**Table 4. Negative binomial regression modelling showing association between GP missed appointment category and number of "did not attends" by specialty category.**

| | Relative risk of 'did not attend' outpatient appointments by specialty category | | |
| --- | --- | --- | --- |
| | Rate ratio (95% CI), p-value | | |
| | Reference group: Zero GP missed appointments | | |
| | **Low** | **Medium** | **High** |
| Adult medicine 'did not attend' | 1.73 (1.70–1.77) p<0.01 | 2.76 (2.69–2.85) p<0.01 | 3.80 (3.68–3.93) p<0.01 |
| Adult surgical services 'did not attend' | 1.60 (1.57–1.64) p<0.01 | 2.35 (2.29–2.41) p<0.01 | 3.03 (2.95–3.12) p<0.01 |
| Maternity/reproductive health 'did not attend | 1.91 (1.83–2.00) p<0.01 | 3.30 (3.14–3.48) p<0.01 | 4.64 (4.36–4.94) p<0.01 |
| Mental health services 'did not attend' | 2.19 (2.09–2.30) p<0.01 | 4.25 (4.02–4.49) p<0.01 | 7.83 (7.35–8.35) p<0.01 |
| Paediatrics 'did not attend' | 1.96 (1.86–2.08) p<0.01 | 3.07 (2.84–3.33) p<0.01 | 3.22 (2.86–3.63) p<0.01 |

Models show rate ratio (RR), 95% Confidence Intervals (CI) and control for age, sex, SIMD, and number of long-term conditions.

Despite this, there appeared to be a clear association between missed GP appointments and both attendance and non-attendance at specific secondary care specialties. Notably, the biggest effects appeared in mental health services, where those who were "high" missers of GP appointments were over four times more likely to attend mental health services and over seven times as likely to miss appointments in mental health services as those who missed no GP appointments. This association is particularly concerning given that GP patients with mental health-based LTCs who were 'high missers' are over eight times more likely to die prematurely compared to those who had mental-health based LTCs but missed no appointments [2].

## Strengths and weaknesses

This dataset represented 1/6 of Scottish citizens and reflected Scotland's population. Using well recognised and robust linkage methods we were able to examine patients use of general practice and hospital care. This paper builds on already published work about the

**Table 5. Negative binomial regression modelling showing association between primary care missed appointment category and number of did not attend appointments/'irregular discharges'/'left before care completed'.**

| | Relative risk of 'did not attend'/'irregular discharge'/'left without care completed' | | |
| --- | --- | --- | --- |
| | Rate ratio (95% CI), p-value | | |
| | Reference group: Zero missed appointments | | |
| | **Low** | **Medium** | **High** |
| Overall outpatient (SMR00) 'did not attend' | 1.13 (1.11–1.14) p<0.01 | 1.33 (1.31–1.34) p<0.01 | 1.62 (1.60–1.64) p<0.01 |
| General hospital admissions (SMR01) 'irregular discharge'* | 1.74 (1.66–1.82) p<0.01 | 2.70 (2.57–2.85) p<0.01 | 4.56 (4.31–4.81) p<0.01 |
| Emergency department (ED) 'left without care completed' | 0.92 (0.88–0.96) p<0.01 | 0.94 (0.89–1.00) p = 0.04 | 1.02 (0.95–1.09) p = 0.61 |

Models show rate ratio (RR), 95% Confidence Intervals and control for age, sex, SIMD, and number of long term conditions. (*numbers too low to model maternity (SMR02) and mental health (SMR04) associations).

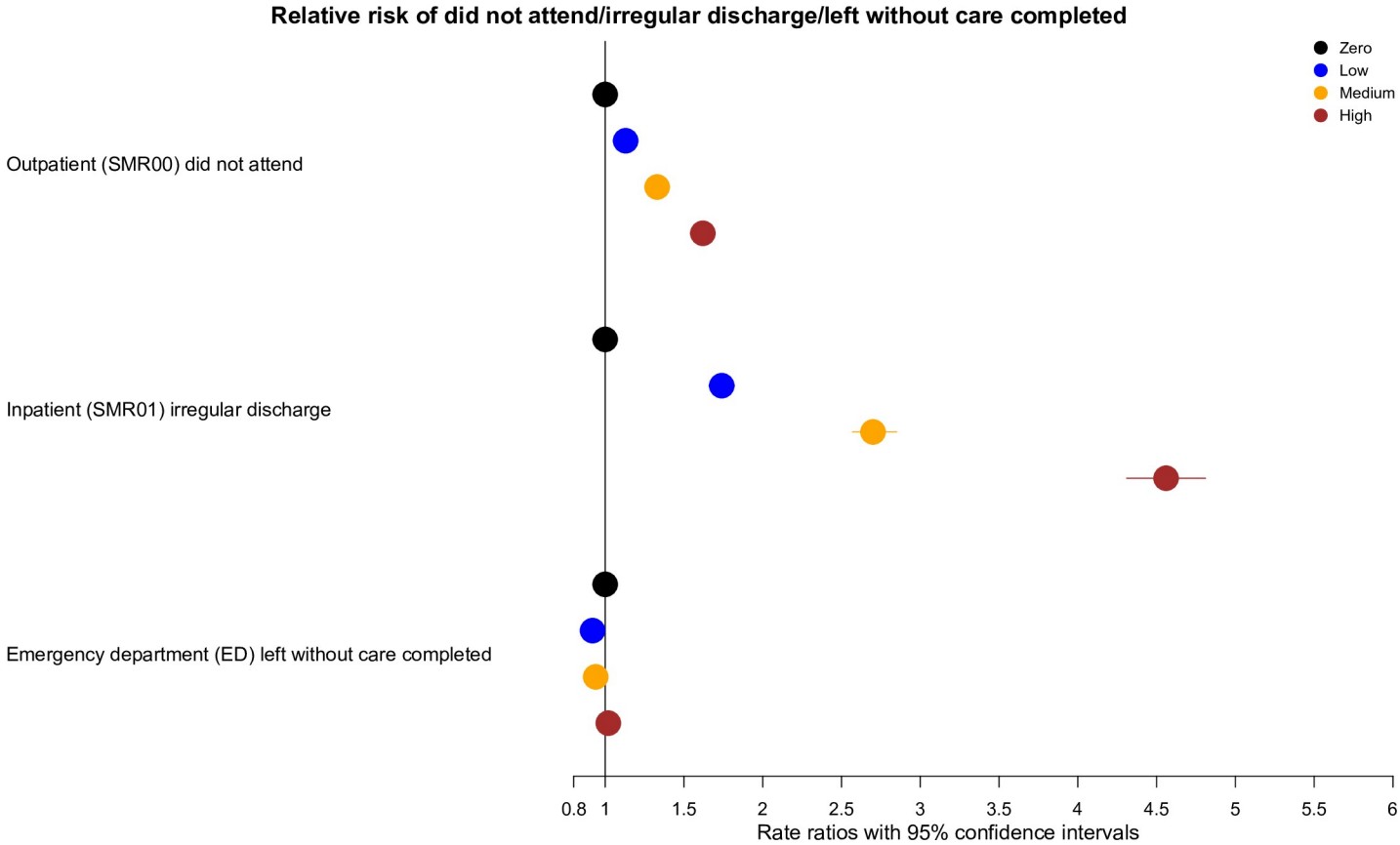

**Fig 2.**

demographics and health outcomes associated with patterns of missed appointments from a large population sample at the patient level, so is able to extend what we know about health care use and 'missingness A weakness is that we were not able to examine the temporal relationship between missed GP appointments and hospital utilisation, and the underlying reasons for non-attendance. Data from NHS 24, Out of hours GP contacts and ambulance call outs would have added more detail to patterns of service use too. We were unable to assess potential temporal associations between GP appointments and hospital activity (including admissions), making it impossible to assess potential causal mechanisms.

We were not able to construct a meaningful rate for missed outpatient appointments which accounted for appointments scheduled in hospital care, like we were for GP missed appointments.

In the UK NHS health care system, almost all citizens are registered with a GP at a single GP practice and patients can schedule general practice appointments as they wish. Our data therefore accurately reflect primary care health service usage. In contrast, hospital service speciality appointments are not scheduled by patients directly and are only possible following a referral by a patient's GP, with a small proportion coming from other hospital specialists or following a hospital admission. Different hospital specialties organise their own appointment scheduling with for example some using letter opt in systems; patients have to follow up and contact the service to arrange their appointment and some will use text message reminders Others will automatically discharge a patient if they do not manage to attend one appointment.

## Strengths and weaknesses in relation to other studies, discussing important differences in results

This research is the first to investigate patterns of missed appointments at the patient level using a large population sample and to examine patients' service use across the health care system. Numerous previous studies have either made no distinction between missing one appointment and many: those that have considered multiple appointments focus on small clinical samples, or single disease areas [1,3].

We were surprised to find no difference in ED attendance between groups. ED attendances across high income countries show a consistent trend of increasing over time, with patients from the most socio-economically deprived communities [13] showing higher rates compared to the general population. One recent UK study which linked GP practice with ED patient level data found that multi-morbidity was the strongest predictor of high ED attendance rather than reduced primary care service access [14]. That our results did not show any of these trends suggests the picture is even more complex.

## Meaning of the study: Possible explanations and implications for clinicians and policymakers

Our findings suggest that patients with higher patterns of missed GP appointments experience higher treatment burden shown by higher rates of outpatient clinic attendances and hospital admissions compared to patients who miss no GP appointments. Lower adherence to treatment plans may also contribute to this. Mental health service missingness is very high and in the context of previously reported mortality outcomes particularly our associations with existing mental health diagnoses; this issue deserves immediate further attention.

It remains to be established whether the patterns of 'missingness' from health care described in our research are associated causally with adverse health outcomes. These findings should however encourage policymakers, health service planners and clinicians to consider the role and contribution that 'missingness' in health care should make to improving patient safety and care [15].

## Unanswered questions and future research

The association between missing multiple health care appointments, high health care use and poor health may be further confounded by other factors including frailty [16], neurodevelopmental problems such as attention-deficit/hyperactivity disorder [17], neurodegenerative disease [18] or psychological trauma [19], in addition to other wider determinants of health (e.g., socioeconomic status). These factors individually or in combination may impact on a person's ability to organise, attend to, or follow through on offers of care and these merit further research, policy and service design attention.

Examination of the relative contribution of multiple factors was not previously possible due to either constraints on available datasets for linkage, or poor quality data [20].

This research helps to highlight the potential benefits that whole system data analysis brings to an understanding of how best to optimise care for patients in health services. However, many questions remain on how new technologies, including apps that let patients link personal data with electronic health records, will best help those in high-risk groups engage with health care.

Being able to predict non-attendance at the patient level may be an important element in providing responsive patient centred care across health care systems. Future research should further examine missingness in health care across community and acute sector services, with a focus on temporality; seeking to understand why these patterns occur at the patient level.

There is now evidence to support the development and evaluation of interventions to reduce non-attendance with a view to assessing impact on morbidity- and mortality-based outcomes with economic analysis. Future work will further elucidate existing potential interventions, assess the role of risk prediction tools and consider tailored relational focused interventions such as care navigators.

## Supporting information

**S1 File. SMR 00 hospital specialty category.**
(DOCX)

## Acknowledgments

Thank you to all the GP practices who participated in this study and for strategic support from Ellen Lynch (Health and Social Care Analytical Services, Scottish Government). The general practice data expertise of Dave Kelly (Albasoft) was invaluable. Thanks also to the eDRIS team who facilitated the safe use of our data in the Safehaven, especially Dionysis Vragkos.

## Author Contributions

**Conceptualization:** Andrea E. Williamson, David A. Ellis, Alex McConnachie, Philip Wilson.

**Formal analysis:** Ross McQueenie, David A. Ellis.

**Funding acquisition:** Andrea E. Williamson, David A. Ellis, Alex McConnachie, Philip Wilson.

**Investigation:** Andrea E. Williamson, Ross McQueenie, David A. Ellis, Philip Wilson.

**Methodology:** David A. Ellis, Philip Wilson.

**Project administration:** Andrea E. Williamson.

**Resources:** Andrea E. Williamson.

**Software:** Ross McQueenie, David A. Ellis.

**Supervision:** Andrea E. Williamson, Ross McQueenie, David A. Ellis, Alex McConnachie.

**Validation:** Alex McConnachie.

**Writing – original draft:** Andrea E. Williamson.

**Writing – review & editing:** Andrea E. Williamson, Ross McQueenie, David A. Ellis, Alex McConnachie, Philip Wilson.

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
