## [Decision Letter · Decision Letter 0]

30 Dec 2020

PONE-D-20-37099

‘Missingness’ in health care: associations between hospital utilization and missed appointments in general practice. A retrospective cohort study.

PLOS ONE

Dear Dr. Williamson,

Thank you for submitting your manuscript to PLOS ONE. After careful consideration, we feel that it has merit but does not fully meet PLOS ONE’s publication criteria as it currently stands. It is a well-written paper that covers an important topic. However, the reviewers highlight several methodological aspects that need to be answered. Therefore, we invite you to submit a revised version of the manuscript that addresses the points raised during the review process.

We look forward to receiving your revised manuscript.

Kind regards,

Juan F. Orueta, MD, PhD

Academic Editor

PLOS ONE

Journal Requirements:

2.We note that you have indicated that data from this study are available upon request. PLOS only allows data to be available upon request if there are legal or ethical restrictions on sharing data publicly. For information on unacceptable data access restrictions, please see http://journals.plos.org/plosone/s/data-availability#loc-unacceptable-data-access-restrictions.

3.Thank you for stating the following in your Competing Interests section: 

"NO"

Reviewers' comments:

Reviewer's Responses to Questions

**Comments to the Author**

1. Is the manuscript technically sound, and do the data support the conclusions?

Reviewer #1: Yes

Reviewer #2: Partly

2. Has the statistical analysis been performed appropriately and rigorously? 

Reviewer #1: Yes

Reviewer #2: No

3. Have the authors made all data underlying the findings in their manuscript fully available?

Reviewer #1: Yes

Reviewer #2: Yes

4. Is the manuscript presented in an intelligible fashion and written in standard English?

Reviewer #1: Yes

Reviewer #2: Yes

5. Review Comments to the Author

Reviewer #1: Review of PONE-D-20-37099 ‘Missingness’ in health care: associations between hospital utilization and missed appointments in general practice. A retrospective cohort study. By Andrea E Williamson et coll.

With much interest and pleasure, I read this good and interesting article. Deviating consultation behavior, including missing appointments, may reveal to us important information about our patients and physician-patients partnership. Hopefully observations of patient’s consultation behavior will lead to changes in our clinical work and the organization of healthcare. Most studied internationally is (persistent) frequent attendance. Frequent attenders have more somatic, psychological and socials problems and give rise to more use of primary and secondary healthcare. Panic and (health) anxiety cause persistence of frequent attendance. Not attending/missing appointments is less studied but certainly intriguing. Did the medical problems of these patients disappear meanwhile or are they too much in panic or too anxious to attend their appointment?

With good reason this article examines in a huge combined (GP and secondary care) database the association between missing GP appointments and (non attending) use of secondary care.

Main remarks:

1. I understand that the authors hypothesise that the association between multiple missed appointments in general practice with increased use of hospital services and missingness from hospital care, may be explained by LTC, SES, age and sex.

-However, hypothesing missed appointments as a deviating consultation behavior, one would be more interested whether there is an association between missing appointments and some specific LTC like panic disorder, (health) anxiety and other social and psychiatric problems (e.g. homelessness? Schizophrenia?) (Page 12) Your GP database probably contains this information. Why didn’t you included this in your analysis and is it possible to add such an analysis?

-Does e.g. the Charlson comorbidity index (CCI) not better reflect the burden of diseases than just counting LTC’s?

-Understanding missing appointments differs according to age category: Missing appointments of the youth can e.g. be explained by self-limiting morbidity, or e.g. depression /addiction which may induce passive behaviour, non-attendance of the frail old can be explained by (aggravation of) existing morbidity and memory problems. Is it possible to perform a sub group analysis of young and old patients?

2. The authors use 4 categories of missed appointments. Why didn’t they use a continuous variable?

Minor remarks:

3. Does the NHS use SMS to alert patients for a scheduled appointment?

4. Abstract: - I miss a (short) description of the ‘why’ (we don’t know whether…. therefore we...)

5. page 14: Please describe what you mean by ‘with refinement of codes included in addiction /mental health categories’. Why, what, wherefore?

6. Line 187. Why using SD here instead of CI?

7. Line 193-195. Normally one doesn’t draw a conclusion in the result section.

8. Result section (e.g. line 210-2010). I prefer you use one, neutral method of reporting in the result section. I prefer the one you use in line 217-220 without mentioning two-fold, 50% more etc. You better mention your interpretation in the conclusion section.

9. Line 229. Primary care appointments ore GP appointments?

10. Line 276. I think it would be very interesting to let us know which psychiatric LTC ‘high missers’ have and which LTC ‘no missers’.

11. Line 290. Perhaps it is good to suggest a future prospective cohort research?

12. Line 306-308. If I understand your results well, you controlled for age, sex, SIMD, and number of long-term conditions. In that respect I think it isn’t strange you don’t find an association.

13. Line 311-313. I find it remarkable that both patients who (persistently) attend the GP frequently and patients with higher patterns of missed GP appointments shown higher rates of outpatient clinic attendances and hospital admissions. So, deviating consultation patterns of the GP seem to predict use of secondary care.

14. Line 331. Prospective cohort research? In ‘unanswered questions…’ you don’t mention research of the possible financial consequences of missing appointments.

15. Tables.

Please explain all abbreviations (CI, ED etc) in a legend.

Please consider to mention the P in a legend and not in the table.

I think you better mention ‘Models show relative risk ratio (RRR) and control for age, sex, SIMD, and number of long-term conditions’ in a legend.

No abbreviations in the title of tables.

Reviewer #2: The authors assembled a retrospective cohort of patients from a nationally representative sample of general practices (GP) in Scotland for examining the association between missed appointment in GP and hospital utilization.

1. The authors used relative risk ratio in text and relative risk in tables to describe the association between missed appoints in GP and hospital utilization. It will be helpful for the authors to use one measure throughout the manuscript. Since the authors used negative binomial model to analyze hospital utilization, I think it is better to use incidence rate ratio (IRR).

2. I disagree with the inclusion of the number of GP appointments scheduled in the model as a covariate. This variable was very likely to be correlated with the main independent variable: missed GP appointments. Instead, the authors can use this variable in defining the main independent variable as a rate (number of missed GP appointments divided by the number of scheduled GP appointments). Zero missed GP appointments means very differently for different numbers of scheduled GP appointments.

3. As the authors pointed out, the temporal association between missed GP appointments and hospital utilization was not assessed. The outcomes and the main independent variables were measured in the same time period. Thus, the current design could not address the question if missed GP appointments caused higher hospital utilization. A different design may help, for example, one can measure missed GP appointments in the first two years and then examine their association with hospital utilization in the third year. Is this or similar approach feasible? If not, please justify why not.

4. I am confused with highlighted texts. Are they authors’ notes or are they what they wanted to emphasize? For example, in line 177, “..advised they agreed…”; in line 208, “additional file 1”.

5. The authors found a strong association between missed GP appointments and attendance of mental health services and missed appointments in mental health services. I am wondering if mental health issues may have caused missed GP appointments. The authors may consider adding prior mental health diagnosis as a covariate in their models.

6. In lines 292-295, it is confusing. The number of scheduled GP appointments was adjusted in models (line 167), not used in defining the rate of missed GP appointments. But here (line 295), the authors said that the number of scheduled GP appointments was used in defining the rate of missed GP appointments.

7. Some minor problems. In line 145, a period was missing; in line 213, change “four-and-a-half” to “4.5” (and other places alike); in line 234, HMA was defined previously, it was not necessary to define it again; between line 292 and 293, “;” was not properly used for separating nouns.

6. PLOS authors have the option to publish the peer review history of their article (what does this mean?). If published, this will include your full peer review and any attached files.

Reviewer #1: **Yes: **Frans T Smits

Reviewer #2: **Yes: **stanley xu

---

## [Author Response · Author response to Decision Letter 0]

16 Mar 2021

This is uploaded as a separate word document.

---

## [Decision Letter · Decision Letter 1]

8 Apr 2021

PONE-D-20-37099R1

‘Missingness’ in health care: associations between hospital utilization and missed appointments in general practice. A retrospective cohort study.

PLOS ONE

Dear Dr. Williamson,

Thank you for submitting your manuscript to PLOS ONE. After careful consideration, we feel that it has merit but does not fully meet PLOS ONE’s publication criteria as it currently stands. Therefore, we invite you to submit a revised version of the manuscript that addresses the points raised during the review process.

Besides the comments of the reviewer, I have other questions.

1. I don’t fully understand the results. The patients in the group with high missed GP appointments (HMA) have higher use of outpatient services (RR=1.46). However, when observing the RR of the 5 categories of outpatient clinics (table-3) all of them are more elevated than 1.46. Something similar is found in the groups of medium and low missed GP appointments. This fact is even more patent in the “did not attend” outpatient appointments. The RRs of any category of the outpatient clinic are higher than the overall RR.2. It is very unusual to mention some limitations of the study in the introduction. Lines 84-91 recognize that observational studies don’t allow to establish causality; lines 93 and following mention several factors that influence the “missingness” of appointments but some of them were not included in the analyses; and lines 105-13 describe differences in hospital specialties organization that can affect the results. Such information is very relevant, but it would be more appropriate to include it in the discussion.3. In lines 193-9 there is a description of results also presented in table-1. However, such description is not very accurate. HMA patients received 2.5 times more outpatient appointments and over 5 times more admissions than those who missed no GP appointments. This effect is even more pronounced in maternity and mental health admissions (19.5 and 8 times, respectively).4. Lines 211-4. The system to classify specialties into 5 categories should be included in the methods section, instead of results.

We look forward to receiving your revised manuscript.

Kind regards,

Juan F. Orueta, MD, PhD

Academic Editor

PLOS ONE

Journal Requirements:

Reviewers' comments:

Reviewer's Responses to Questions

**Comments to the Author**

1. If the authors have adequately addressed your comments raised in a previous round of review and you feel that this manuscript is now acceptable for publication, you may indicate that here to bypass the “Comments to the Author” section, enter your conflict of interest statement in the “Confidential to Editor” section, and submit your "Accept" recommendation.

Reviewer #1: (No Response)

Reviewer #2: All comments have been addressed

2. Is the manuscript technically sound, and do the data support the conclusions?

Reviewer #1: Yes

Reviewer #2: Yes

3. Has the statistical analysis been performed appropriately and rigorously? 

Reviewer #1: Yes

Reviewer #2: Yes

4. Have the authors made all data underlying the findings in their manuscript fully available?

Reviewer #1: Yes

Reviewer #2: Yes

5. Is the manuscript presented in an intelligible fashion and written in standard English?

Reviewer #1: Yes

Reviewer #2: Yes

6. Review Comments to the Author

Reviewer #1: Review of PONE-D-20-37099R1

Full Title: ‘Missingness’ in health care: associations between hospital utilization and missed

appointments in general practice. A retrospective cohort study.

I want to thank the authors for their elaborate answers. I think they have certainly clarified many issues and improved their article.

However, they were not able to solve some issues due to (legal and database) limitations. I think it would be wise to mention this in ‘limitations’.

- You state that it is not possible to investigate the association between specific LTC’s (especially psychological ones like anxiety and panic) and missingness. I think you have to mention this in ‘limitations’. Also, why this wasn’t possible.

- You agree that missingness probably is very different for the (very) old and the young with (perhaps) very different drivers, but it was not possible to perform a subgroup analysis. Please mention this in the discussion section (limitations).

In the discussion section I miss some words about possible (clinical) explanations of the associations you found. Personally, I find it remarkable that not only patients who miss GP-consultations, but also patients who frequently attend the GP, use more specialist care. As a GP, I think there might be common underlying issues like (health) anxiety, panic and personality. Your article gains persuasiveness if you explore this briefly in the discussion section. (‘Meaning of the study…’)

Only knowledge about these possible causal clinical and social factors, can lead to useful interventions. Please mention which interventions you have in mind. (page 16 line 338)

Minor point: In table 2,3, and 4 you state ‘SIMD’. However, in table 5 you state the full description.

Reviewer #2: The authors included the number of GP appointments as an offset in negative binomial model offset. The data were available from NHS Scotland. Permission for access was granted to the study team only, from the participating GP practices and the Public Benefit and Privacy Panel NHS Scotland. I have no further questions. Thanks for addressing my concerns.

7. PLOS authors have the option to publish the peer review history of their article (what does this mean?). If published, this will include your full peer review and any attached files.

Reviewer #1: **Yes: **Frans T. Smits

Reviewer #2: **Yes: **Stanley Xu

---

## [Editor Report · Decision Letter 2]

1 Jun 2021

‘Missingness’ in health care: associations between hospital utilization and missed appointments in general practice. A retrospective cohort study.

PONE-D-20-37099R2

Dear Dr. Williamson,

We’re pleased to inform you that your manuscript has been judged scientifically suitable for publication and will be formally accepted for publication once it meets all outstanding technical requirements.

Kind regards,

Juan F. Orueta, MD, PhD

Academic Editor

PLOS ONE

---

## [Editor Report · Acceptance letter]

15 Jun 2021

PONE-D-20-37099R2 

‘Missingness’ in health care: associations between hospital utilization and missed appointments in general practice. A retrospective cohort study. 

Dear Dr. Williamson:

I'm pleased to inform you that your manuscript has been deemed suitable for publication in PLOS ONE. Congratulations! Your manuscript is now with our production department. 

Kind regards, 

on behalf of

Dr. Juan F. Orueta 

Academic Editor

PLOS ONE